# Research on Weigh-in-Motion Algorithm of Vehicles Based on BSO-BP

**DOI:** 10.3390/s22062109

**Published:** 2022-03-09

**Authors:** Suan Xu, Xing Chen, Yaqiong Fu, Hongwei Xu, Kaixing Hong

**Affiliations:** School of Mechanical and Electrical Engineering, China Jiliang University, Hangzhou 310018, China; p1901085204@cjlu.edu.cn (X.C.); fuyaqiong@cjlu.edu.cn (Y.F.); xhw@cjlu.edu.cn (H.X.); hongkaixing@cjlu.edu.cn (K.H.)

**Keywords:** WIM, BSO algorithm, wavelet transform, BP neural network

## Abstract

Weigh-in-motion (WIM) systems are used to measure the weight of moving vehicles. Aiming at the problem of low accuracy of the WIM system, this paper proposes a WIM model based on the beetle swarm optimization (BSO) algorithm and the error back propagation (BP) neural network. Firstly, the structure and principle of the WIM system used in this paper are analyzed. Secondly, the WIM signal is denoised and reconstructed by wavelet transform. Then, a BP neural network model optimized by BSO algorithm is established to process the WIM signal. Finally, the predictive ability of BP neural network models optimized by different algorithms are compared and conclusions are drawn. The experimental results show that the BSO-BP WIM model has fast convergence speed, high accuracy, the relative error of the maximum gross weight is 1.41%, and the relative error of the maximum axle weight is 6.69%.

## 1. Introduction

The problem of vehicle overload seriously endangers traffic safety, not only easily causing traffic accidents, but also causing fatal damage to the structure of roads and bridges [1]. Weight measurement is the most direct way to identify overloaded vehicles. Static weighing is a very mature method for weighing stationary vehicles with an accuracy of 0.1%. However, static weighing of moving vehicles on the road requires the vehicle to stop, which is likely to cause traffic jams and affect the efficiency of vehicle traffic. Compared with static weighing, weigh-in-motion (WIM) can measure the weight of a moving vehicle without affecting vehicle traffic, so it has a higher detection efficiency. However, the application of WIM is limited due to its low precision. A proper estimation of weighing accuracy is crucial for WIM systems to be efficient in direct enforcement of overloading [2]. The study of Lhoussaine Oubrich et al. shows that the accuracy of a WIM system is mainly affected by the quality of the road surface, axle load, speed, and suspension system structure [3]. In addition, the distance between the sensor and the wheel, the installation depth position of the sensor, and the pavement temperature also affect the weighing accuracy of the WIM system [4,5]. There are many factors that affect the accuracy of WIM, and the relationship between different influencing factors is unknown. Therefore, it is difficult to determine the relationship between the accuracy of WIM and different factors. Artificial neural networks (ANN) possess the characters of strong nonlinear mapping, adaptive learning, fault-tolerance performance, and robustness. They are widely used in pattern recognition, data processing, fault diagnosis, and so on. Some researchers have used neural networks to improve the accuracy of WIM systems. Sungkon Kim et al. used ANN to analyze the WIM signal of the main girders and the weighing information of the vehicle to calculate the vehicle weight and axle weight of the vehicle. Their study shows that the accuracy of ANN method is higher than that of the influence line method in axle load calculation [6]. Yun Zhou et al. used a deep convolutional neural network to distinguish vehicle weight and bridge structure response to estimate vehicle attributes [7]. Zhixin Jia et al. used the BP model to identify the sensor closest to the tire–pavement contact area, and then used the signal from the sensor to predict the vehicle weight. In their experiments, 96.5% of the test samples had a relative error of less than 5% [8]. However, the BP neural network is sensitive to the initial weights and can easily fall into the local optimum, which makes its performance unable to reach the optimum. Optimization algorithms such as the swarm intelligence algorithm and genetic algorithm (GA) can adjust the initial weight of the BP neural network and reduce the possibility of falling into the local optimum. Tiantian Wang et al. compared the performance of BSO, particle swarm optimization (PSO), GA, and the grasshopper optimization algorithm (GOA) on 23 benchmark functions. Experiments show that the performance of BSO algorithm is better than other algorithms [9]. The BSO algorithm combines the information sharing mechanism of the PSO algorithm and the search mechanism of the beetle antennae search (BAS) algorithm is applied in different fields [10]. Jianming Zhou et al. used the BP neural network optimized by the BSO algorithm to predict the parameters of each part of the twisted pair and improved the prediction accuracy of the crosstalk of the twisted pair [11]. Lei Wang adjusted the control parameters of the robot hand trajectory planning through the BSO algorithm and compared it with the PSO algorithm and the genetic algorithm to verify the superiority of the BSO algorithm [12]. F.N. Al-Wesabi et al. used BSO to optimize the weights and bias parameters of the least squares support vector machine to achieve a better classification effect [13]. Parminder Singh et al. proposed an adaptive neuro-fuzzy inference system for heart disease and multi-disease diagnosis and used the BSO algorithm to optimize the parameters of the inference system, which improved the accuracy and precision of diagnosis [14]. Aiming at the defects of the BP neural network and the superiority of the BSO algorithm, this paper proposes a WIM model based on the BSO algorithm and BP neural network. Firstly, the structure and principle of the WIM system used in this paper are analyzed. Secondly, the WIM signal is denoised and reconstructed by wavelet transform. Then, a BP neural network model optimized by the BSO algorithm is established to process the WIM signal. Finally, the predictive ability of BP neural network models optimized by different algorithms are compared and conclusions are drawn. The purpose of this paper is to study the effect of improving the weighing accuracy of dynamic weighing system based on a BP neural network optimized by different algorithms.

The paper is organized as follows:Structure and principle of the WIM system.Wavelet transform algorithm.BSO-BP algorithm.Experimental results and analysis.Conclusions.

## 2. Structure and Principle of the WIM System

The components of the WIM system include an integrated weighing platform, embedded strain sensor, signal processing circuit, ground sensing coil, industrial computer, and cloud platform. A structural diagram of a WIM system is shown in Figure 1. When the vehicle passes the weighing platform, the embedded strain sensor deforms under pressure and converts the weight signal into the voltage signal. The signal processing circuit converts the voltage signal into the digital signal. The industrial computer calculates parameters such as vehicle weight, vehicle speed, and number of axles, and uploads these parameters to the cloud platform via the network.

The test platform is built on the road and the weighing platform is shown in Figure 2.

Two weighing platforms are installed in one lane. The two platforms are side by side. A single weighing platform has a horizontal width of 1.75 m and a longitudinal width of 0.8 m. The bottom of the weighing platform is supported by two symmetrically placed I-shaped beams. Two pressure sensors are installed symmetrically on an I-shaped beam. The structure diagram of the weighing platform is shown in Figure 3.

## 3. Pre-Processing of the WIM Signal

Wavelet transform is a transform analysis method which is widely used in the fields of fault detection, image processing, and signal analysis [15,16,17,18,19]. Since the WIM signal is a discrete signal, this paper uses discrete wavelet transform to process the WIM signal.

Daubechies (*dbN*) wavelet function is sensitive to irregular signals and it is widely used in signal analysis. *N* is the order of the *dbN* wavelet function. When *N* is equal to 4, the *dbN* wavelet has the characteristics of good orthogonality and large vanishing moment. Multi-layer decomposition and reconstruction of a weighing signal by *db4* wavelet can reduce the interference of noise signal and obtain a real WIM signal.

The waveforms before and after the wavelet transform are shown in Figure 4 and Figure 5, respectively.

According to Figure 4 and Figure 5, it can be seen that the high-frequency noise in the waveform is significantly reduced. The results show that the wavelet transform reduces the interference of noise and makes the waveform closer to the ideal waveform.

## 4. BSO-BP Algorithm

### 4.1. BP Neural Network

The BP neural network has strong nonlinear mapping ability, which is widely used in the fields of pattern recognition, parameter estimation, motion control, and classification [20,21,22,23,24]. The structure of the BP neural network is shown in Figure 6.

In Figure 6, *X_t_* (1 < *t* < *n*) is the input nodes and *Y_u_* (1 < *u* < *m*) is the output nodes. In addition, *w_ij_* represents the weight of the input layer to the hidden layer and *w_jk_* represents the weight of the hidden layer to the output layer.

### 4.2. PSO Algorithm

The PSO algorithm can search the best region in complex space through group cooperation [25]. Each particle constantly adjusts its search behavior by learning from its own experience and that of other particles. Particles update their velocity and position by the following formulas during each iteration:V_i_(k + 1) = ωV_i_(k) + c_1_r_1_[p_ibest_ − X_i_(k)] + c_2_r_2_[p_gbest_ − X_i_(k)](1)
X_i_(k + 1) = X_i_(k) + V_i_(k)(2)
where V_i_ is the velocity of the particle i, X_i_ is the position of the particle i, p_ibest_ is the historical optimal position of the particle i, p_gbest_ is the global optimal position, k is an evolution algebra, ω is an inertia weight, c_1_ and c_2_ represent learning factors, and r_1_ and r_2_ represent random numbers between (0,1). In addition, the position and speed of the particles need to be limited by parameters.

### 4.3. Principle of BAS and BSO Algorithms

The BAS algorithm imitates the beetle’s search mechanism and random behavior [26]. The beetle uses its left and right antennae to detect the strength of food smells and adjusts the direction of its search. Based on this simple principle, it can easily find food. Yinyan Zhang et al. studied the convergence of the BAS algorithm, and the results show that BAS algorithm has good performance [27]. The steps of the BAS algorithm are as follows:Create a normalized random vector b. The calculation formula is as follows:
(3)$$b=\frac{\mathrm{rands}\left(m,1\right)}{\left\|\mathrm{rands}\left(m,1\right)\right\|}$$
where rands() is a random function, m is a spatial dimension, and b represents the vector from the left antennae to the right antennae;
2.Create a spatial search model for the left and right antennae:
x_rk_ = x^k^ + d^k^ × b/2(4)
x_lk_ = x^k^ − d^k^ × b/2(5)
where x_rk_ represents the position coordinates of the right antennae at the kth iteration, x_lk_ represents the position coordinates of the left antennae at the kth iteration, x^k^ represents the centroid coordinates of the beetle at the kth iteration, and d^k^ represents the search distance between the left and right antennae at the kth iteration;
3.f(x_lk_) and f(x_rk_) are calculated using the fitness function f(x);4.Iteratively update the position of the beetle through search behavior:
(6)$$x^{k+1}=x^{k}-\delta^{k}\times b\times \mathrm{sign}\left(f\left(x_{rk}\right)-f\left(x_{lk}\right)\right)$$
where δ^k^ represents the step size of the tth iteration, and sign() represents the sign function;
5.Update search distance and step size:
d^k^ = η_d_ × d^k^^−1^ + d_0_(7)
δ^k^ = η_δ_ × δ^k^^−1^ + δ_0_(8)
where η_d_ is is the attenuation coefficient of the search distance, η_δ_ is the attenuation coefficient of the step size, and d_0_ and δ_0_ represent the minimum distance threshold and the minimum step threshold. Set d_0_ and δ_0_ to avoid the value of d^k^ and δ^k^ being 0.

The objective function is calculated only twice in each iteration, so the BAS algorithm has strong global search ability [28]. However, the single search characteristic of the BAS algorithm makes it easy to fall into the local optimum when processing high-dimensional data, and it can only solve the single-objective optimization problem [29]. The BSO algorithm combines the search method of the BAS algorithm and the group information sharing mechanism of the PSO algorithm, which not only accelerates iterative convergence, but also reduces the possibility of falling into local optimum [30]. In the BSO algorithm, the random direction b in the expression of the antennae is replaced by the speed of the PSO algorithm. The expression is as follows:x^k^_ird_ = x^k^_id_ + v^k^_id_ × d^k^/2(9)
x^k^_ild_ = x^k^_id_ − v^k^_id_ × d^k^/2(10)
where x^k^_ild_ and x^k^_ird_ are the position vectors of the left and right antennae of particle i in the dth dimension at the kth iteration.

Increment of searching behavior of beetle antennae:ξ^k+1^_id_ = δ^k^ × v^k^_id_ × sign(f(x^k^_ird_) − f(x^k^_ild_))(11)
where ξ^k+1^_id_ represents the increment of the search behavior at the (k + 1)th iteration.

The updated formula for the position and velocity of the beetle swarm is as follows:v^k+1^_id_ = ωv^k^_id_ + c_1_r_1_(pbest^k^_id_ − x^k^_id_) + c_2_r_2_(gbest^k^_id_ − x^k^_id_)(12)
x^k+1^_id_ = x^k^_id_ + v^k+1^_id_ + ξ^k+1^_id_(13)
where x^k^_id_ represents the position of the beetle i at the kth iteration, x^k+1^_id_ represents the speed of the beetle i at the (k + 1)th iteration, c_1_ and c_2_ represent learning factors, r_1_ and r_2_ represent random numbers between (0,1), pbest^k^_id_ is the historically optimal position of the beetle i, gbest^k^_id_ represents the historical global optimal position of the beetle swarm.

In the PSO algorithm, inertia weight ω is a fixed value. The research shows that the inertia weight has a great influence on the searching range of particles. This paper introduces the inverted S-shaped function to adjust the inertia weight to improve the searching ability of the algorithm. The updated formula of the inertia weight is as follows:(14)$$\omega =\omega_{\max}-\frac{\omega_{\max}-\omega_{\min}}{1+e^{a-bt}}$$
where ω represent random numbers between (0.4,0.9), b is equal to 0.2, and a is equal to 5.

### 4.4. Establishment of the BSO-BP WIM Model

The random initial weight of the BP neural network makes the network unstable and easily falls into local optimum. The BSO algorithm is used to optimize the weights and thresholds of the BP neural network, which can reduce the possibility of the network falling into a local optimum. The optimization steps are as follows:The data processed by wavelet transform is used as the input sample of BSO-BP neural network;Set the size of the beetle swarm, the maximum number of iterations, the inertia weight, and the search space dimension, etc. The search space dimension is calculated as follows:
D = m × n + r × n + n + r(15)
where m represents the number of neurons in the input layer, n represents the number of neurons in the hidden layer, and r represents the number of neurons in the output layer;

3.Randomly generate the position and speed of the beetle. According to formula 25, the fitness function value is calculated. Save the individual optimal value and the group optimal value:(16)$$f\left(x_{i}\right)=\frac{1}{N}\sum_{i=1}^{N}{\left(x_{i}^{\prime }-x_{i}\right)}^{2}$$
where N represents the number of training samples, x′_i_ represents the predicted value of the training sample, and x_i_ represents the true value of the training samples;
4.Update the step size of the beetle:
δ^k+1^ = eta × δ^k^(17)
where k is the number of iterations and eta is equal to 0.95.
5.Iterative optimization. Iteratively update the position, velocity and inertia weight of the beetle. The individual optimal value of the beetle and the group optimal value of the beetle swarm are updated according to the fitness value of the beetles;6.The final iteration result is taken as the initial weight and threshold value of BP neural network;7.Train the BSO-BP WIM model. Continuously update the weight and threshold of the network according to the error until the set accuracy is reached.

## 5. Experimental Results and Analysis

A total of three vehicles were tested. Among them, there is one two-axle car, one four-axle car, and one six-axle car. In addition, each vehicle is divided into two states, empty and full, to simulate different cargo loading conditions. Before the test, three vehicles were statically weighed to obtain the static gross weight and static axle weight. After the test, the dynamic gross weight, dynamic axle weight, and vehicle speed of the vehicle are recorded. The test vehicles pass as close to a constant speed as possible, and the vehicle speed is less than 70 km/h. A total of 261 groups of experimental data were collected. According to the experimental data, the gross weight dataset and the axle weight dataset were respectively composed. The gross weight dataset includes vehicle speed, number of axles, dynamic gross weight, and static gross weight. The axle weight dataset includes vehicle speed, number of axles, axle number, dynamic axle weight, and static axle weight. The parameters of all tested vehicles are shown in Table 1.

### 5.1. Data Pre-Processing

This paper used MATLAB to construct a BSO-BP WIM model. The input data and output data were normalized using the mapminmax function to improve the training speed of the samples.

After the training of the BSO-BP WIM model, the prediction results of the model were processed by inverse normalization.

### 5.2. Establishment of WIM Model and Parameter Selection

The input layer has three nodes, namely dynamic vehicle weight, vehicle speed, and axle number. The purelin is the transfer function from the hidden layer to the output layer. In the BSO algorithm, the population size *N* is 10, the learning factors *c*_1_ and *c*_2_ are both equal to 2, and the convergence accuracy *E* is 0.001. The training time of the BP network is 20,000, the learning rate is 0.01, and the target error is 0.000004. The gross weight test set and the gross weight training set are randomly selected from the gross weight data set. Among them, 10% of the data is used as the gross weight test set, and the remaining 90% of the data is used as the gross weight training set for model training. Axle weight test set and axle weight training set are established in the same way.

### 5.3. Prediction of Gross Vehicle Weight and Analysis of Results

The BSO-BP WIM model, the BP WIM model, and the PSO-BP WIM model are established through the gross weight training set. The prediction errors of the three models on the gross weight test set are compared. The comparison result is shown in Figure 7.

In Figure 7, the gross weight prediction error of the BSO-BP WIM model is significantly lower than that of the BP WIM model and the PSO-BP WIM model.

The average gross weight relative error and the maximum gross weight relative error of the three models are shown in Table 2.

As can be seen from Table 2, the average gross weight relative error and maximum gross weight relative error of the BSO-BP WIM model are lower than those of the PSO-BP WIM model and the BP WIM model. The mean relative error of the gross weight test set is reduced from 5.58% to 0.53%, and the maximum relative error is reduced from 15.07% to 1.41%. The results show that the BSO algorithm improves the accuracy and generalization ability of the BP neural network, and its optimization effect is higher than that of the PSO algorithm.

The fitness function values of the PSO-BP neural network and BSO-BP neural network are compared. The fitness function value is the absolute value of the difference between the predicted gross weight and the static gross weight. The comparison result is shown in Figure 8.

In Figure 8, the PSO-BP algorithm needs more than 50 iterations to reach the optimum and its fitness value is about 70. The BSO-BP algorithm only needs 25 iterations, and its fitness value is about 20. The results show that the BSO algorithm can improve the convergence speed, enhance the optimization ability, and prevent the algorithm from falling into local optimum.

### 5.4. Prediction of Axle Weight and Analysis of Results

The BSO-BP WIM model, the BP WIM model, and the PSO-BP WIM model are established through the axle weight training set. The prediction errors of the three models on the axle weight test set were compared. The comparison results are shown in Figure 9.

The average axle weight relative error and the maximum axle weight relative error of the three models are shown in Table 3.

As can be seen from Table 3, the axle weight prediction error of the BSO-BP WIM model is smaller than that of other models. The mean relative error of the axle weight test set is reduced from 3.38% to 1.61%, and the maximum relative error is reduced from 6.86% to 6.69%. From the comparison results of Table 2 and Table 3, it can be seen that the BSO algorithm has a higher prediction accuracy for gross weight or axle weight.

Three axle weight samples with different axle numbers were selected from the axle weight dataset. The original waveforms of the three axle weight samples were corrected according to the predicted axle load. The corrected waveforms, original waveforms, and static waveforms of the three axle weight samples were compared, respectively. The comparison results are shown in Figure 10, Figure 11 and Figure 12.

As can be seen from Figure 10, Figure 11 and Figure 12, the predicted axle weight of the BSO-BP WIM model is the closest to the static axle weight. The comparison of axle weight waveforms is more intuitive, which further verifies that the BSO algorithm has better performance.

## 6. Conclusions

This paper proposed a BP WIM model optimized based on the BSO algorithm to improve the accuracy of the WIM system. The BSO algorithm combines the information sharing mechanism of the PSO algorithm and the single search mechanism of the BAS algorithm and has a stronger ability to find capabilities. After the initial weights of the BP neural network are optimized by the BSO algorithm, the generalization and convergence capabilities of the network are improved. After the optimization of the BSO-BP model, the average relative error of axle weight of test set was reduced from 3.38% to 1.41%, the maximum relative error of axle weight was reduced from 6.86% to 6.69%, the average relative error of gross weight was reduced from 5.89% to 0.53%, and the maximum relative error of gross weight was reduced from 15.07% to 1.41%. Experimental results show that BSO-BP model can improve the accuracy of the WIM system, and the prediction accuracy of gross weight was higher than that of axle weight. Meanwhile, the prediction accuracy of gross weight and axle weight using the BSO-BP model is higher than that using the PSO-BP and BP models. The results verify the superiority of the BSO algorithm, indicating that the optimization ability of the algorithm is stronger than that of the PSO algorithm.

Although the BSO algorithm can improve the accuracy of the dynamic weighing system, its performance is easily affected by parameters, and it takes more time to adjust the parameters of the model to achieve better results. In addition, limited by the experimental conditions, the number of test vehicles is small, and the actual vehicle traffic situation cannot be completely simulated.

## Figures and Tables

**Figure 1 sensors-22-02109-f001:**
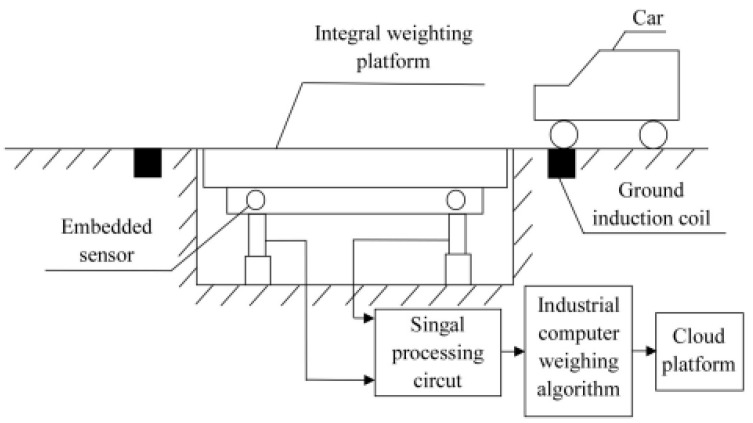
Structural diagram of a WIM system.

**Figure 2 sensors-22-02109-f002:**
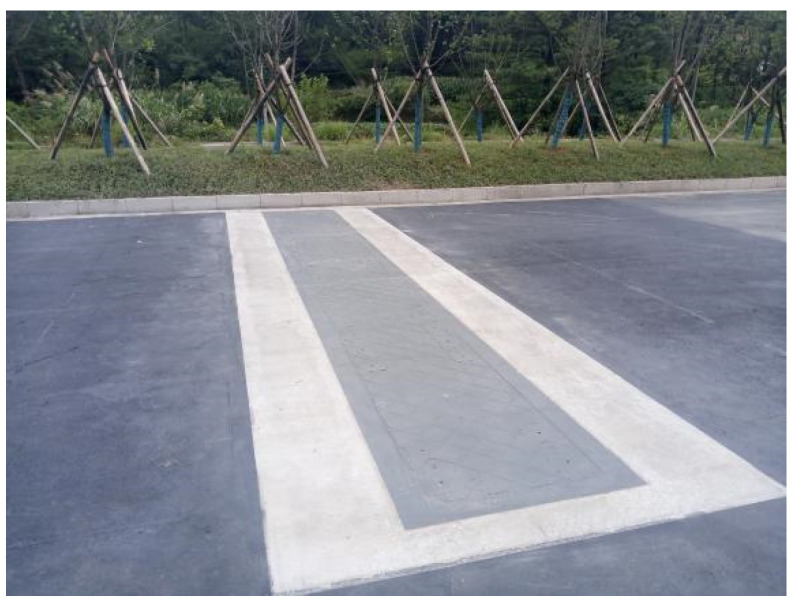
Weighing platform.

**Figure 3 sensors-22-02109-f003:**
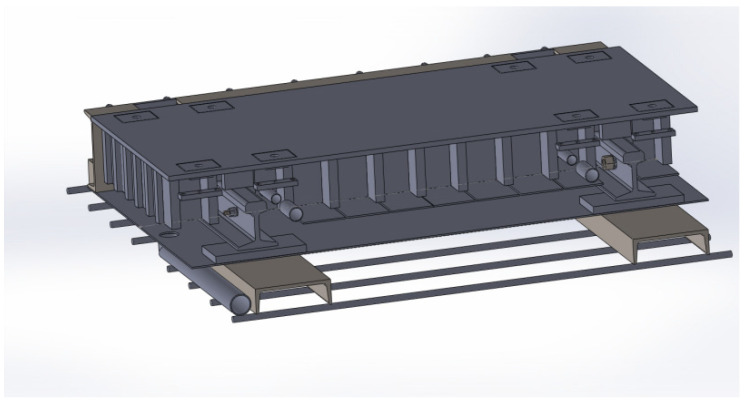
Schematic diagram of the scale structure.

**Figure 4 sensors-22-02109-f004:**
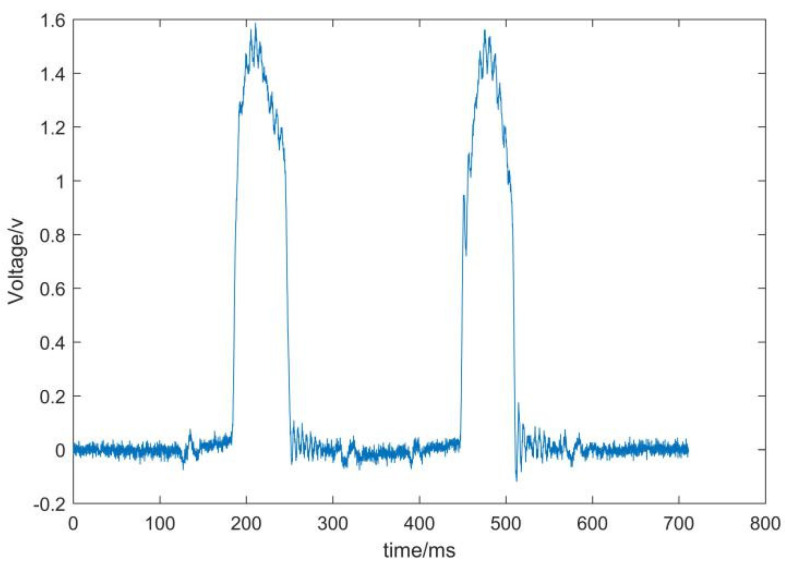
Weighing signal without wavelet transform.

**Figure 5 sensors-22-02109-f005:**
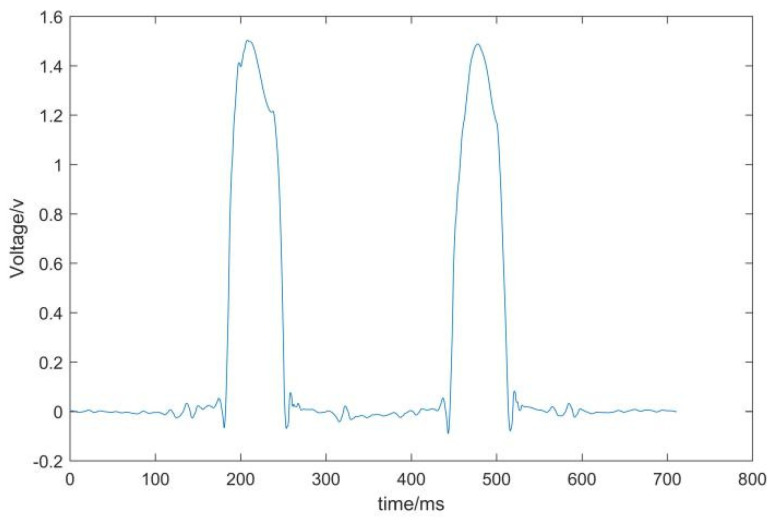
The weighing signal after wavelet transform.

**Figure 6 sensors-22-02109-f006:**
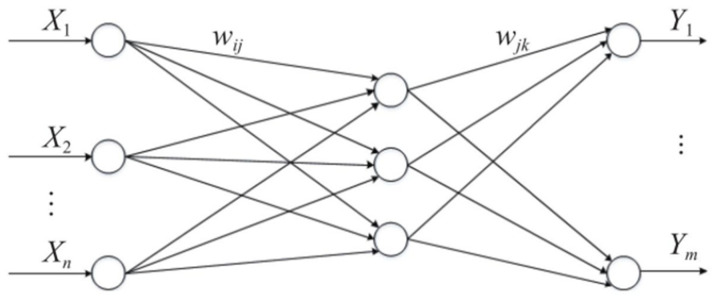
Structure diagram of the BP neural network.

**Figure 7 sensors-22-02109-f007:**
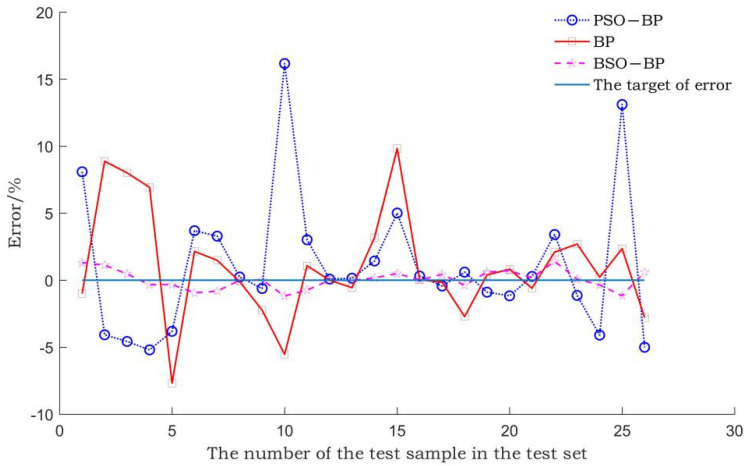
Comparison of gross weight prediction errors of different models.

**Figure 8 sensors-22-02109-f008:**
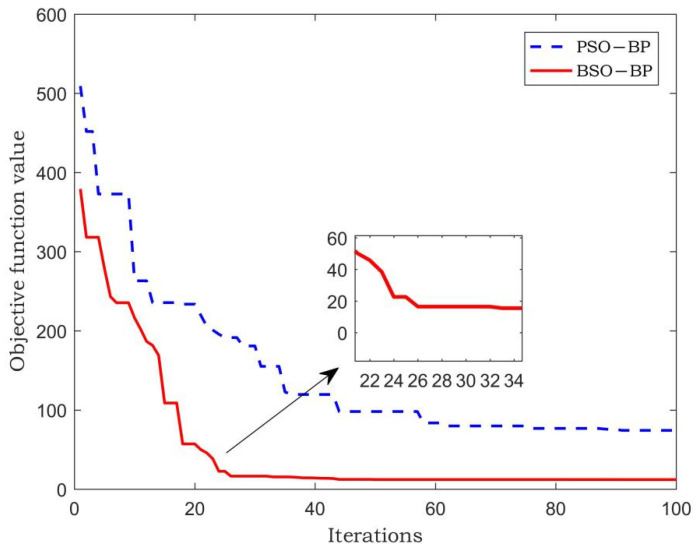
Iterative comparison of fitness function values for gross weight.

**Figure 9 sensors-22-02109-f009:**
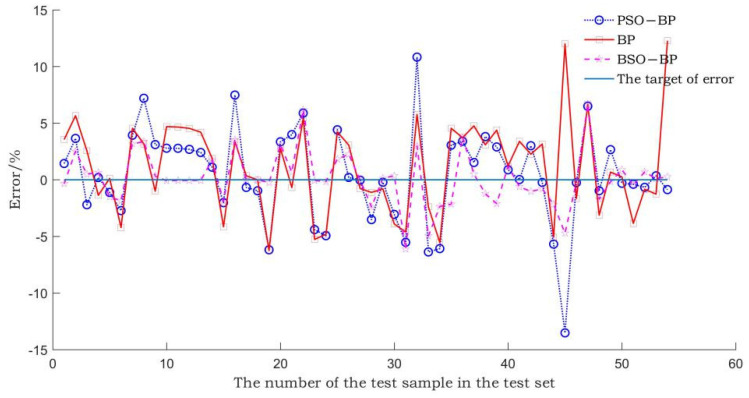
Comparison of axle weight prediction errors of different models.

**Figure 10 sensors-22-02109-f010:**
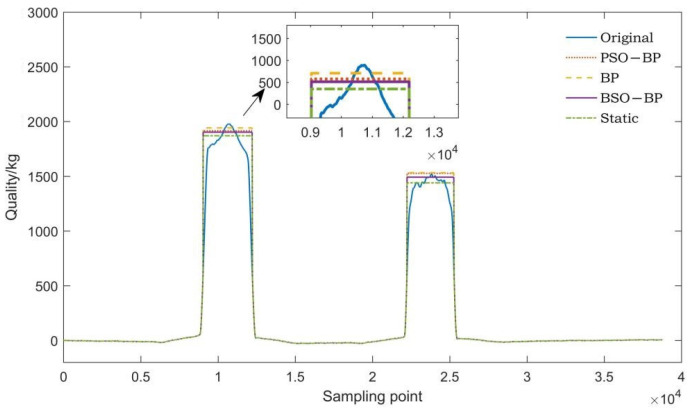
Corrected axle weight waveform comparison of a two-axle vehicle.

**Figure 11 sensors-22-02109-f011:**
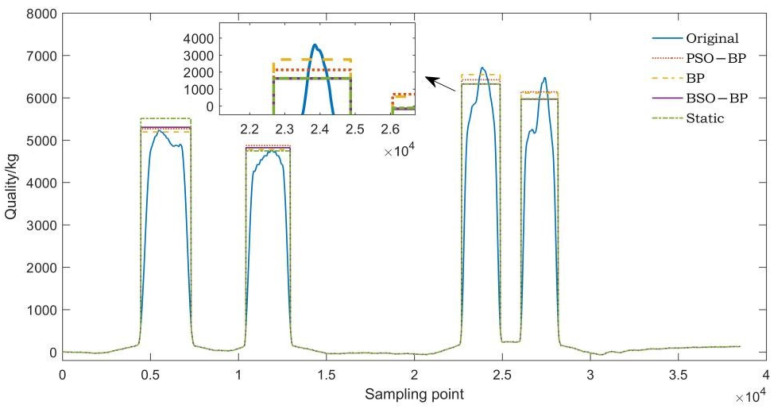
Corrected axle weight waveform comparison of a four-axle vehicle.

**Figure 12 sensors-22-02109-f012:**
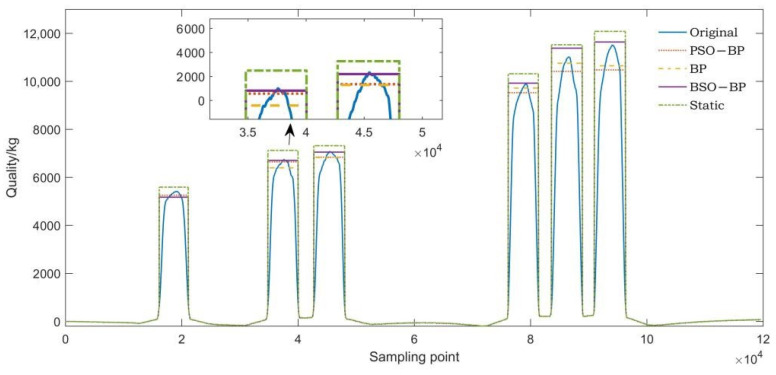
Corrected axle weight waveform comparison of a six-axle vehicle.

**Table 1 sensors-22-02109-t001:** Data from three test vehicles.

Number of Axles	Load	First Axle (kg)	Second Axle (kg)	Third Axle (kg)	Fourth Axle (kg)	Fifth Axle (kg)	Sixth Axle (kg)	Static Weight (kg)
2	Empty	1870	1440	/	/	/	/	3310
	Full	2358	2932	/	/	/	/	5290
4	Empty	5512	4746	6326	5976	/	/	22,560
	Full	7339	8001	24,756	25,434	/	/	65,530
6	Empty	4847	2905	2981	1197	1823	2957	16,710
	Full	5591	7123	7318	10,315	11,523	12,080	53,950

**Table 2 sensors-22-02109-t002:** Mean gross weight relative error and maximum gross weight relative error for the three models.

Model	Mean Gross Weight Relative Error/%	Maximum Gross Weight Relative Error/%
BP	2.83	9.82
PSO-BP	2.62	18.78
BSO-BP	0.53	1.41

**Table 3 sensors-22-02109-t003:** Mean axle weight relative error and maximum axle weight relative error of three models.

Model	Mean Axle Weight Relative Error/%	Maximum Axle Weight Relative Error/%
BP	3.51	12.28
PSO-BP	3.12	13.52
BSO-BP	1.61	6.69

## Data Availability

The data presented in this study are available in Appendix A.

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
