# Peer review of "Research on Weigh-in-Motion Algorithm of Vehicles Based on BSO-BP"

_sensors, 2022, doi:10.3390/s22062109_

Round 1

Reviewer 1 Report

the manuscript is well organized, the subject matter is very current.
it is presented in a clear and legible way, it can be considered a good scientific work.
I would advise sutors to augment the introductory part by making a reference to self-driving vehicles. means of transport (car; trains; planes).
the conclusions must be increased they are superficial they must be valued.
check the english language. check the format of the magazine.
increase the bibliography.

There is more work that needs to be done in terms of :

1.Presentation of the  research. 2. Clarity of the research methodology.

Besides, I suggest the following bibliographical references;

Decision Tree Method to Analyze the Performance of Lane Support Systems

G Pappalardo, S Cafiso, A Di Graziano, A Severino

Sustainability 13 (2), 84

Reviewer 2 Report

This paper proposes the method that use beetle swarm algorithm and error back propagation neural network to realize weigh-in-motion. The method is verified by a simulation case. The paper organization must be improved to make the articles better display the research results. In the revised edition, the author must put forward some opinions.

  1. Please organize the narration of the introduction. The innovation of the proposed method should be emphasized.
  2. It is mentioned in line 111 that wavelet transform reduces noise interference and makes the waveform closer to the ideal waveform. However, the difference between the waveforms in Fig. 4 and Fig. 5 is not obvious.
  3. In the simulation case (line 204), how is the data measurement and working condition simulated under different vehicle and loading conditions? More details of the operation of simulation should be provided.
  4. Please provide relevant experimental verification or engineering cases if possible.

Reviewer 3 Report

In the present paper a beetle swarm optimization (BSO) algorithm and the error back propagation (BP) neural network applied in a Weight in Motion model is presented.

The overall algorithm is clearly presented.

Since you are using measurements you should take into consideration the uncertainty of the measurements.

In the Simulation results and analysis section the weight per axle for the 4-axle vehicle should be checked again since they seem high (according to typical technical ability of the axles).

Please define the way experimental data are created and give more details. How many of them are related to the 2-axle, the 4-axle and the 6-axle vehicle? A table like Table 1 should be created from experimental dataset with the average values of weight for each case of load and each vehicle. Is there any repeatability in the experimental set? How does the vehicle speed affect the measurements?

Do the percentages of errors in Tables 2 & 3 change versus the number of the axles of the vehicle?

Please elaborate on how the sampling sequence affect your results.

Figures 10 - 12 would be better to be presented according to the weight (kg) and not to the sensor measurements (Voltage/V). This way also the static weight can be presented in the same graph. 

Correct the final conclusion (6.69% refers to the axle weight)

Round 2

Reviewer 2 Report

The reviewer has no opinion on the paper, and suggest that the paper can be accepted and published.